# COVID-19 during pregnancy and risk of pregnancy loss (miscarriage or stillbirth): a systematic review protocol

Jennifer Campbell ![ORCID],[1,2] Rachael Williams,[2] Mia Harley,[1] Krishnan Bhaskaran[1]

[1]NCDE, London School of Hygiene & Tropical Medicine Faculty of Epidemiology and Population Health, London, UK
[2]Clinical Practice Research Datalink, Medicines and Healthcare Products Regulatory Agency, London, UK

**Correspondence to**
Jennifer Campbell;
jennifer.campbell@mhra.gov.uk

## ABSTRACT

**Introduction** The COVID-19 pandemic has led to concerns about potential adverse pregnancy outcomes associated with infection, resulting in intensive research. Numerous studies have attempted to examine whether COVID-19 is associated with an increased risk of pregnancy loss. However, studies and reviews to date have drawn differing conclusions. The aim of this systematic review is to provide a summary of all quantitative research on the relationship between pregnancy loss and COVID-19 infection and, if appropriate, to synthesise the evidence into an overall effect estimate.

**Methods and analysis** Three publication databases (Embase, PubMed and Cochrane) and four preprint databases (medRxiv, Lancet Preprint, Gates Open Research and Wellcome Open Research) will be searched. Boolean logic will be used to combine terms associated with pregnancy loss and COVID-19. The population of interest are pregnant women. Retrieved results will be assessed in two phases: (1) abstract screening and (2) full text evaluation. All studies which compare pregnancy loss outcomes in women who had COVID-19 versus those who did not quantitatively will be included. Narrative and non-English studies will be excluded. Two reviewers will screen independently, with results compared and discrepancies resolved by the study team. Study quality and risk of bias will be assessed using a quality appraisal tool. Results will be summarised descriptively and where possible synthesised in a meta-analysis.

**Ethics and dissemination** This systematic review requires no ethical approval. This review will be published in a peer-reviewed journal and provide an important update in a rapidly evolving field of research.

**PROSPERO registration number** CRD42022327437.

## STRENGTHS AND LIMITATIONS OF THIS STUDY

⇒ This systematic review will include both published and preprint studies in an attempt to capture the very latest data and minimise publication bias.
⇒ Study selection, data extraction and quality assessment will be performed independently by two researchers, which will ensure that all relevant studies are included without personal biases.
⇒ All included studies will be assessed for quality using the National Institute for Health and Care Excellence quality appraisal checklist for quantitative studies reporting correlations and associations.
⇒ Studies which are not published in English will not be included. This limitation may cause language bias.

## BACKGROUND

SARS-CoV-2 emerged as a new coronavirus at the end of 2019 spreading rapidly to cause a global pandemic of its associated illness COVID-19. Many millions of people around the world have been infected with the virus including pregnant women. However, due to the novelty of COVID-19 little is known about its potential effect on the unborn fetus and pregnancy outcomes. Aetiological hypotheses have been proposed as to ways in which COVID-19 may adversely affect pregnancy outcomes including potential increased risk of loss mediated by placental damage.[1] COVID-19 in pregnancy has therefore been the subject of intense research and there have been numerous studies which have examined any potential adverse effect leading to reviews which have attempted to summarise the evidence.[2–4]

As both the virus itself and our knowledge of its effects are constantly evolving both studies and reviews to date have drawn differing conclusions. Some have concluded an increased risk of pregnancy loss associated with COVID-19 infection[2 5–9] while others have concluded no increased risk.[10–13] Many early reviews of this question included only case reports as no comparative studies were available.[4 14–16] The latest published systematic review on this question by Pathirathna *et al* included studies published prior to June 2021 just over 1 year into the COVID-19 pandemic and like all reviews to date on this topic they noted the need for further research.[8] Since this review, there have been numerous additional studies published and there has been a global roll-out of vaccinations for COVID-19 to pregnant women. It is therefore important that we continue to review all emerging evidence in order to provide a full and current picture of any potential adverse risk.

**Table 1** Inclusion and exclusion criteria

| Inclusion | Exclusion |
|---|---|
| ► Epidemiological studies which attempt to quantitatively assess any association between pregnancy loss and COVID-19. (Study designs may include prospective and retrospective cohort studies, case–control studies and cross-sectional studies.) | ► Non-English language publications including those where the summary is in English but not the full text.<br>► Narrative review articles, guidelines, editorials or comments.<br>► Studies without a control or comparison group, for example, case reports.<br>► Conference presentations. |

The overall aim of this study is to identify and summarise all studies to date which have quantitatively compared pregnancy loss outcomes in women who contracted COVID-19 while pregnant versus those who did not. Where possible, quantitative estimates of associations between COVID-19 and pregnancy loss will be synthesised into an overall effect estimate.

## METHODS
### Study registration
This protocol is prepared in accordance with the Preferred Reporting Items for Systematic Review and Meta-Analysis Protocols statement (online supplemental appendix 1).[17] This protocol is registered on the International Prospective Register of Systematic Reviews (PROSPERO; registration number: CRD42022327437).

### Eligibility criteria
The review will include all studies which have attempted to quantitatively assess the potential association between having COVID-19 during pregnancy and pregnancy loss.

The population of interest are pregnant women at any maternal age or gestation of pregnancy. The exposure of interest will be COVID-19 during pregnancy. We will include all studies which attempt to ascertain COVID-19 exposure in pregnancy regardless of the method of diagnosis. The comparator population will be women who did not have COVID-19 during pregnancy. The outcome of interest will be pregnancy loss (miscarriage or stillbirth). Table 1 gives the inclusion and exclusion criteria that will be applied to identified studies.

### Information sources
Publication databases to be searched: Embase (Ovid), PubMed, Cochrane.

**Table 2** Database search strategy

| Database | Dates of search coverage | Miscarriage/stillbirth | COVID-19 |
|---|---|---|---|
| PubMed | 1 March 2020 to current date | 'Abortion, Spontaneous' [MeSH] OR 'Fetal Death' [MeSH] OR 'Stillbirth' [MeSH] OR (miscarriage [MeSH Terms])) OR (miscarriages [MeSH Terms] OR Miscarriage* OR pregnancy loss* OR spontaneous abortion* OR fetal loss* OR foetal loss* OR foetal death* OR fetal death* | 'coronavirus' [MeSH] OR 'coronavirus infections' [MeSH Terms] OR 'coronavirus' [All Fields] OR 'covid 2019' [All Fields] OR 'SARS2' [All Fields] OR 'SARS-CoV-2' [All Fields] OR 'SARS-CoV-19' [All Fields] OR 'severe acute respiratory syndrome coronavirus 2' [supplementary concept] OR 'coronavirus infection' [All Fields] OR 'severe acute respiratory pneumonia outbreak' [All Fields] OR 'novel cov' [All Fields] OR '2019ncov' [All Fields] OR 'sars cov2' [All Fields] OR 'cov22' [All Fields] OR 'ncov' [All Fields] OR 'covid19' [All Fields] OR 'covid 19' [All Fields] OR 'covid-19' [All Fields] OR 'coronaviridae' [All Fields] OR 'corona virus' [All Fields] |
| Embase | 1 March 2020 to current date | spontaneous abortion/exp OR stillbirth/exp OR stillbirth.m.p OR pregnancy loss/exp OR pregnancy loss.mp OR foetal death.m.p OR fetus death OR fetus death/exp NOT [medline]/lim | 'coronavirinae'/exp OR 'coronavirinae' OR 'coronaviridae infection'/exp OR 'coronaviridae infection' OR 'coronavirus disease 2019'/exp OR 'coronavirus'/exp OR coronavirus OR 'coronavirus infection'/de NOT [medline]/lim |
| Cochrane | 1 March 2020 to current date | Search for 'stillbirth' OR 'miscarriage' OR 'foetal death rates' OR 'foetal death rate' OR 'fetal death' OR 'fetal death rate' OR 'pregnancy loss rate' OR 'pregnancy loss-rate' OR pregnancy 'loss-rates' | Search for 'coronavirus' in the Title Abstract Keyword fields |

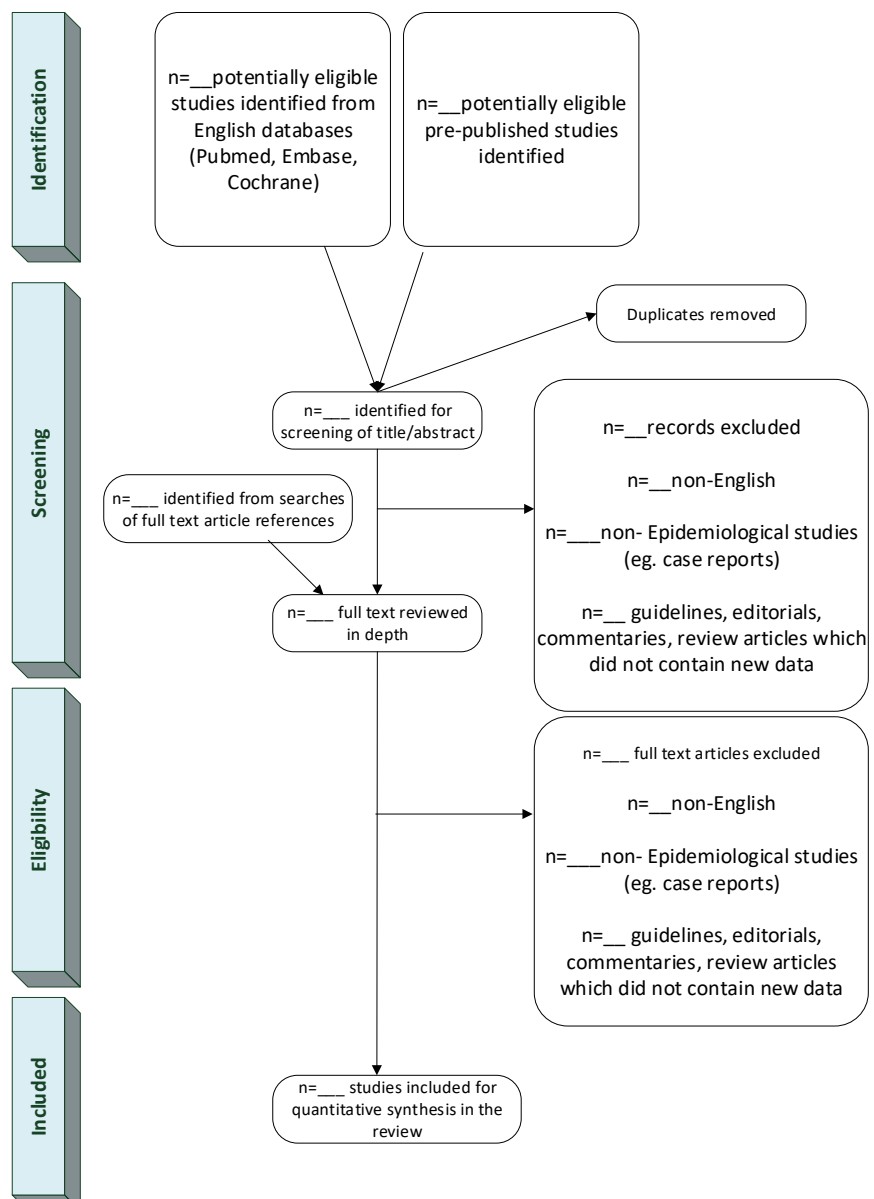

**Figure 1** Preferred Reporting Items for Systematic Reviews and Meta-Analyses (PRISMA) diagram of study selection process.

Preprint platforms to be searched: medRxiv, Lancet Preprint, Gates Open Research, Wellcome Open Research.

### Search strategy

Search terms listed in table 2 will be applied in the respective databases. Terms related to pregnancy loss will be combined with terms related to COVID-19 using AND logic. Only publications after 1 March 2020 will be searched.

To further increase the sensitivity of our search, the list of references from review articles relating to COVID-19 and pregnancy loss will be screened manually to identify other potentially eligible articles.

Due to the fast-moving nature of COVID-19 research we will also search databases of preprint articles.[18] The medRxiv database will be searched via Embase using the search terms detailed above. The Lancet Preprint database will be searched for Obstetrics and Gynaecology articles which contain the term 'Covid-19'. Gates Open Research and Wellcome Open Research will also be searched for 'Covid-19' and 'Pregnancy'. Preprint databases were selected from a systematic examination of preprint platforms by Kirkham *et al.*[19] Preprint articles will be flagged as such in any presentation of results.

### Data management and selection process

Searches will be performed across all databases by reviewer 1. Records of the search terms, results from the search and the date of last run will be saved. Results will be exported into Mendeley where any duplicate results will be removed.[20] Each article will be given a study ID. The remaining articles will be screened for eligibility based on titles and abstracts by two independent reviewers applying the inclusion/exclusion criteria described

**Table 3** Example of data collection form

| Study ID | First author, year | Study design | Location | Exposure definition | Outcome definition | Subjects (n) | Exposed (n) | Miscarriage among the exposed (n) | Stillbirth among the exposed (n) | Miscarriage among the unexposed (n) | Stillbirth among the unexposed (n) | Statistical measure and result reported in the paper | Was the study before or after vaccine roll-out? |
|---|---|---|---|---|---|---|---|---|---|---|---|---|---|
| - | | | | | | | | | | | | | |

above. Discrepancies will be discussed and, where necessary, will be decided by the whole study team. Full text articles will be obtained for all articles deemed eligible for inclusion from the initial screening. Articles will be divided and assessed independently by two reviewers after which the final selection will be agreed. Any reasons for exclusion will be recorded. The study selection process is outlined in figure 1.

### Data collection process

The example data capture form (table 3) will be pilot tested on a random sample of five included studies and revised if necessary. The finalised data capture form will then be completed by reviewers 1 and 2 independently for a sample of 10 studies to check concordance, after which each study will be examined by one reviewer.

### Assessment of study quality

All included studies will be assessed for bias by reviewers using an adapted version of the National Institute for Health and Care Excellence (NICE) quality appraisal checklist for quantitative studies reporting correlations and associations (online supplemental appendix 2).[21] The NICE tool was chosen as it is designed for identifying rigour in observational studies that explore and generate hypotheses about causal relationships and can be used for multiple study designs. The NICE tool consists of five major items: study population and participants; selection and methods; outcomes; analysis; and summary.

Appraisal will be done using an Excel format to allow for easy compilation of responses. Decisions will be discussed and any discrepancies resolved. Each study will then be awarded an overall study quality grade for external and internal validity from one of the three categories below which are based on the checklist criteria (online supplemental appendix 1).

► ++All or most of the checklist criteria have been fulfilled, where they have not been fulfilled the conclusions are very unlikely to alter.
► +Some of the checklist criteria have been fulfilled, where they have not been fulfilled, or not adequately described, the conclusions are unlikely to alter.
► − Few or no checklist criteria have been fulfilled and the conclusions are likely or very likely to alter.

Studies deemed to be low quality (category) will be excluded from any meta-analysis.

### Data synthesis

We will use Higgins and Thompson's $I^2$ statistic to quantify heterogeneity, and if $I^2$ is >50% meta-analysis will be conducted in Stata using a random-effects model.[22] Where meta-analysis is attempted funnel plots will be used to assess publication bias.[23] Where statistical pooling is not possible, findings will be presented in narrative form using tables to aid in data presentation. If possible, we will conduct subgroup analyses of studies reporting miscarriage and stillbirth separately. We will also look at any potential impact of the widespread use of COVID-19 vaccines by grouping studies into those conducted before and after vaccine roll-out if possible. We will use 1 March 2021 as the cut-off date for studies considered to be post-vaccine roll-out. For studies after this date we will examine the national vaccine roll-out programme for the country in which the study was conducted to assess the likelihood that pregnant women within the study would have been vaccinated. We will also consider a subgroup analysis of hospitalised versus non-hospitalised COVID-19 cases if there are enough studies which consider this.

### Patient and public involvement

There will be no patient or public involvement in this project.

## DISCUSSION

The COVID-19 pandemic has been a challenging time for pregnant women, knowledge on the potential risks of infection to them and their unborn babies is ever evolving. With COVID-19 now circulating widely in many countries and limited risk reduction measures in place it is important to try and fully understand the risks so that pregnant women can be advised appropriately. Reviews and studies to date on whether COVID-19 increases the risk of pregnancy loss have drawn mixed conclusions.[2–4 8 13] COVID-19 research is a fast-moving area; therefore, it is important that reviews are regularly updated. This systematic review aims to provide a comprehensive overview of the latest evidence.

COVID-19 research moves very quickly, and preprint literature has become a key outlet for new research with many researchers opting to make their work available as quickly as possible. Including prepublications in this review, something which previous reviews have not done, will allow us to obtain as current a picture as possible of all

of the evidence. Inclusion of preprint literature may also help mitigate any risk of publication bias.

Vaccination against COVID-19 became widely available globally in 2021.[24] In the UK, pregnant women have been routinely advised to receive COVID-19 vaccination together with the rest of the population, according to their age and underlying health conditions since 16 April 2021.[25] The widespread introduction of COVID-19 vaccination may have led to a decrease in potential risk or pregnancy loss. We hope to identify enough studies to allow us to examine separately those which were conducted before and after the vaccination roll-out in order to provide an insight into any impact the vaccine may have had.

The results of this review can be used to inform public health messaging for pregnant women around the potential risks of COVID-19 infection. This research will also help inform any future research studies planned on this question.

**Contributors** This protocol was written by JC with KB, RW and MH performing critical review. JC will act as the guarantor of the review.

**Funding** This work is funded by Wellcome (Senior Research Fellowship for KB, grant number: 220283/Z/20/Z). This work forms part of JC's PhD which is funded by the Clinical Practice Research Datalink, a division of the UK Medicines and Healthcare Products Regulatory Agency.

**Competing interests** None declared.

**Patient and public involvement** Patients and/or the public were not involved in the design, or conduct, or reporting, or dissemination plans of this research.

**Patient consent for publication** Not applicable.

**Provenance and peer review** Not commissioned; externally peer reviewed.

**ORCID iD**
Jennifer Campbell http://orcid.org/0000-0002-0684-4437

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
