## [Reviewer comments · BMJ Open]

ARTICLE DETAILS

TITLE (PROVISIONAL)	Covid-19 during pregnancy and risk of pregnancy loss (miscarriage or stillbirth): a systematic review protocol
AUTHORS	Campbell, Jennifer; Williams, Rachael; Harley, Mia; Bhaskaran, Krishnan

VERSION 1 – REVIEW

REVIEWER	Cai, Jianghui Chengdu Women's and Children's Central Hospital
REVIEW RETURNED	01-Jul-2022

GENERAL COMMENTS	I have read the manuscript "Covid-19 during pregnancy and risk of pregnancy loss (miscarriage or stillbirth): a systematic review protocol", submitted to BMJ Open. In this paper, the authors aimed to investigate the relationship between pregnancy loss and Covid-19 infection. I have a few comments which I think will improve the paper, which is quite well written. "The review will include all studies which have attempted to quantitatively assess the potential association between having Covid-19 during pregnancy and pregnancy loss." in the Eligibility Criteria section. The authors should clearly state the definition of "having Covid-19 during pregnancy". Can you clarify the method of diagnosis? The COVID-19 was confirmed by clinical or laboratory test? Did the laboratory test include reverse transcriptase-polymerase chain reaction (RT-PCR) and/or antibody testing SARS-CoV-2?
---

REVIEWER	Townson, Julia Cardiff University, Centre for Trials Research
REVIEW RETURNED	26-Aug-2022

GENERAL COMMENTS	1) I wonder if the authors could add how they propose handling different levels of infection within pregnancy, for example those hospitalised compared to those that were not.2) Could the authors specify whether looking at the roll out of vaccination against Covid-19 and risk of stillbirth or miscarriage will be confined to UK studies only?3) Two minor typos:-a. Page 1 line 48 , Narrative and non-English...rather than noneb. Page 10 line 13, I think the word vaccination is missing from the sentence "The widespread introduction of Covid 19..."
--

VERSION 1 – AUTHOR RESPONSE

Reviewer Reports:

Reviewer: 1

Dr. Jianghui Cai, Chengdu Women's and Children's Central Hospital

Comments to the Author:

I have read the manuscript "Covid-19 during pregnancy and risk of pregnancy loss (miscarriage or stillbirth): a systematic review protocol", submitted to BMJ Open.

In this paper, the authors aimed to investigate the relationship between pregnancy loss and Covid-19 infection.

I have a few comments which I think will improve the paper, which is quite well written.

“The review will include all studies which have attempted to quantitatively assess the potential association between having Covid-19 during pregnancy and pregnancy loss.” in the Eligibility Criteria section.

The authors should clearly state the definition of “having Covid-19 during pregnancy”. Can you clarify the method of diagnosis? The COVID-19 was confirmed by clinical or laboratory test? Did the laboratory test include reverse transcriptase-polymerase chain reaction (RT-PCR) and/or antibody testing SARS-CoV-2?

Thank-you for your helpful comments. We will not restrict studies to those which used a certain method to diagnose Covid-19 however, we will extract this information from each included study and group them in our analysis where appropriate. We have added the following to the Eligibility Criteria section of the protocol in order to clarify this:

Changes:

“We will include all studies which attempt to ascertain Covid-19 exposure in pregnancy regardless of the method of diagnosis” (Page 3, Eligibility Criteria Section)

Reviewer: 2

Dr. Julia Townson, Cardiff University

Thank-you for taking the time to review the paper and for your useful comments

Comments to the Author:

1) I wonder if the authors could add how they propose handling different levels of infection within pregnancy, for example those hospitalised compared to those that were not.

We have added hospitalised versus non hospitalised Covid-19 patients as a potential sub-group analysis.

Changes:

“We will also consider a subgroup analysis of hospitalised versus non-hospitalised Covid-19 cases if there are enough studies which consider this.” (Page 8, Data Synthesis Section)

2) Could the authors specify whether looking at the roll out of vaccination against Covid-19 and risk of stillbirth or miscarriage will be confined to UK studies only?

We will consider vaccination status for all studies where possible, not just UK based studies. We have added text to explain this more clearly in the paper

Changes:

"We will use 01/03/2021 as the cut-off date for studies considered to be post vaccine roll-out. For studies after this date we will examine the national vaccine rollout programme for the country in which the study was conducted to assess the likelihood that pregnant women within the study would have been vaccinated." (Page 8, Data Synthesis Section)

3) Two minor typos:-

a. Page 1 line 48 , Narrative and non-English...rather than none b. Page 10 line 13, I think the word vaccination is missing from the sentence "The widespread introduction of Covid 19..."

We have corrected this.